# Inter- and Intra-Observer Agreement of PD-L1 SP142 Scoring in Breast Carcinoma—A Large Multi-Institutional International Study

**DOI:** 10.3390/cancers15051511

**Published:** 2023-02-28

**Authors:** Mohamed Zaakouk, Mieke Van Bockstal, Christine Galant, Grace Callagy, Elena Provenzano, Roger Hunt, Corrado D’Arrigo, Nahla M. Badr, Brendan O’Sullivan, Jane Starczynski, Bruce Tanchel, Yasmeen Mir, Paul Lewis, Abeer M. Shaaban

**Affiliations:** 1Institute of Cancer and Genomic Sciences, University of Birmingham, Birmingham B15 2TT, UK; 2Cancer Pathology, National Cancer Institue, Cairo University, Cairo 12613, Egypt; 3Department of Pathology, Cliniques Universitaires Saint-Luc Bruxelles, 1200 Brussels, Belgium; 4Institut de Recherche Expérimentale et Clinique, Université Catholique de Louvain, 1348 Brussels, Belgium; 5Discipline of Pathology, School of Medicine, Lambe Institute for Translational Research, University of Galway, H91 TK33 Galway, Ireland; 6NIHR Cambridge Biomedical Research Centre, Cambridge CB2 0QQ, UK; 7Addenbrookes Hospital, Cambridge CB2 0QQ, UK; 8Department of Histopathology, Cambridge University NHS Foundation Trust, Cambridge CB2 0QQ, UK; 9Department of Histopathology, Wythenshawe Hospital, Manchester M23 9LT, UK; 10Poundbury Cancer Institute, Dorchester DT1 3BJ, UK; 11Department of Pathology, Faculty of Medicine, Menoufia University, Shebin El-Kom 32952, Egypt; 12Cellular Pathology, Queen Elizabeth Hospital Birmingham, Birmingham B15 2GW, UK; 13Cellular Pathology, Heart of England NHS Foundation Trust, Birmingham B9 5ST, UK; 14Pathology, Royal Liverpool and Broadgreen University Hospitals, Liverpool L7 8YE, UK; 15Medical School, Swansea University, Singleton Park, Swansea SA2 8PP, UK

**Keywords:** PD-L1, breast cancer, VENTANA SP142, triple-negative

## Abstract

**Simple Summary:**

PD-L1 analysis in TNBC is essential for selecting patients eligible for immunotherapy. Limited data are available on pathologists’ concordance regarding PD-L1 assessment. Twelve pathologists of various expertise from three European countries digitally analysed 100 breast cancer core biopsies stained using the SP142 PD-L1 assay in two rounds. The overall inter-observer agreement among the pathologists was substantial. The intra-observer agreement was substantial to almost perfect. The expert scorers were more concordant in evaluating staining percentage compared with those of the non-experts. Challenging cases around the 1% cut-off value for positivity were identified and represented a small 6–8%) proportion of all cases. The experts were more concordant in scoring those cases. The study shows reassuringly strong inter- and intra-observer concordance among pathologists in PD-L1 scoring. A proportion of low-expressors remain challenging to assess, and these would benefit from addressing the technical issues, testing a different sample and/or referring for expert opinions.

**Abstract:**

The assessment of PD-L1 expression in TNBC is a prerequisite for selecting patients for immunotherapy. The accurate assessment of PD-L1 is pivotal, but the data suggest poor reproducibility. A total of 100 core biopsies were stained using the VENTANA Roche SP142 assay, scanned and scored by 12 pathologists. Absolute agreement, consensus scoring, Cohen’s Kappa and intraclass correlation coefficient (ICC) were assessed. A second scoring round after a washout period to assess intra-observer agreement was carried out. Absolute agreement occurred in 52% and 60% of cases in the first and second round, respectively. Overall agreement was substantial (Kappa 0.654–0.655) and higher for expert pathologists, particularly on scoring TNBC (6.00 vs. 0.568 in the second round). The intra-observer agreement was substantial to almost perfect (Kappa: 0.667–0.956), regardless of PD-L1 scoring experience. The expert scorers were more concordant in evaluating staining percentage compared with the non-experienced scorers (R^2^ = 0.920 vs. 0.890). Discordance predominantly occurred in low-expressing cases around the 1% value. Some technical reasons contributed to the discordance. The study shows reassuringly strong inter- and intra-observer concordance among pathologists in PD-L1 scoring. A proportion of low-expressors remain challenging to assess, and these would benefit from addressing the technical issues, testing a different sample and/or referring for expert opinions.

## 1. Introduction

Advances in biomarker assessment, companion diagnostics and genomics have revolutionised the way breast cancer is currently classified and managed [1,2]. The immune microenvironment of solid tumours, including in breast cancer, plays a pivotal role in tumour development and progression [3,4,5]. Cancer cells can evade the regulatory pathways of Programmed death-1 (PD-1) and its ligand (PD-L1), thus overcoming the cytotoxic effect of T cells. Immune checkpoint blockades using anti-PD-L1 inhibitors have been investigated in various trials in lung, melanoma and, more recently, breast cancer, with confirmed efficacy [6,7,8]. This has led to the approval of immune modulators for the treatment of PD-L1-positive breast cancer, and this is currently being incorporated in various guidelines [9]. The first-approved and most established immune checkpoint inhibitor in breast cancer is atezolizumab, for which a companion diagnostic assay (the VENTANA SP142) is required for selecting patients eligible for this drug.

The limited data available in the literature on non-breast cancer suggest the poor reproducibility of PD-L1 SP142 scoring [10]. Some studies compared the performance of various PD-L1 assays [11,12], and only few analysed pathologist concordance in the scoring of breast cancer [13,14,15]. Those latter studies were small and heterogeneous, with some including training sets [14]. Furthermore, the nature of discordant cases was not analysed, nor was there an assessment of the intra-observer agreement or the effect of the pathologist’s experience. In addition, all previous studies focused on TNBC, and, therefore, information on pathologist concordance in the scoring of PD- L1 in HER2-positive and/or luminal breast cancer does not exist. Emerging data suggest cross-talk between HER2 and PD-L1 and potentially support the use of immunotherapy in HER2-positive breast cancer [16]. PD-L1 expression is correlated with the response to neoadjuvant chemotherapy in HER2-positive breast cancer [17].

We therefore aimed to assess the inter- and intra-observer concordance of breast pathologists of various expertise and geographical locations in reporting a large cohort of PD-L1 SP142-stained invasive breast carcinomas of various molecular subtypes to assess if particular molecular subtypes would be more or less prone to poor inter-observer concordance. We also sought to analyse discordant cases in detail to gain insight into the reasons for discrepancies in PD-L1 results, allowing for a subsequent search for strategies on how to tackle them.

## 2. Materials and Methods

Core biopsies from a total of 100 cases of primary breast cancers were included in the study. Cases were selected retrospectively from the files of a single large UK institution (Queen Elizabeth Hospital Birmingham) to include all molecular subtypes with enrichment for the TNBC group.

First, 4 μm sections of formalin-fixed, paraffin-embedded tumour blocks were cut and stained using the VENTANA SP142 anti-PD-L1 rabbit monoclonal primary antibody and a VENTANA Benchmark ULTRA automated staining platform, according to the manufacturer’s protocol. A section from a cell block containing three cell lines with various staining intensities and a section of normal tonsil were included as on-slide controls. Paired H&E sections and PD-L1-stained immunohistochemistry slides were digitally scanned using a Leica Aperio AT2 slide scanner at x40 and uploaded to the University of Birmingham digital platform via a secure link: https://eslidepath.bham.ac.uk, Last accessed 23 February 2023. Each participant was provided with a unique username and password to allow for access to the digital platform for whole slide scoring. Twelve pathologists from eight institutions representing three European countries (United Kingdom, Republic of Ireland, Belgium) evaluated all cases in round one, of whom 10 re-scored the same cases in round two, separated by at least 3 months of a washout period designed to assess intra-observer variability. All pathologists had previously received Roche training for SP142 PD-L1 scoring in TNBC and passed a proficiency test.

PD-L1 SP142 Immune cell (IC) scoring was conducted according to the recommended scoring algorithm [18], using a cut-off value of ≥1% to indicate positivity. In addition, the pathologists were asked to provide their percentage of immune cells with positive staining for each case, including those cases scored as negative. All scorers completed a survey assessing their experience in breast pathology reporting as well as their training and real-life reporting of PD-L1.

### Statistical Analysis

The data were tabulated and statistically analysed using the SPSS (IBMS) software version 28. We used standard statistical analyses for assessing intra/inter-rater concordance/agreement, which have been previously described [19]. Intraclass correlation coefficient (ICC), which is a measure of the reliability of ratings (using median percentage scores), was used to determine if subjects/items can be rated reliably by different raters. ICC is a descriptive statistic used to assess the consistency or reproducibility of quantitative measurements made by different observers measuring the same quantity. The value of an ICC can range from 0 to 1, with 0 indicating no reliability among raters and 1 indicating perfect reliability among raters. The ICC results are interpreted as follows: values < 0.5 indicate poor reliability, values from 0.5 to 0.75 indicate moderate reliability, values from 0.75 to 0.9 indicate good reliability and values greater than 0.9 indicate excellent reliability [20]. In our study, we used a Two-Way Random model, testing both the consistency and the absolute agreement relationships and the mean of ratings as the unit of measurement.

Fleiss multiple-rater Kappa statistics of inter-observer and intra-observer agreement for designating cases as PD-L1-positive versus -negative using a cut-off value of 1% were calculated. Fleiss’ Kappa κ is a measure of inter-rater agreement used to determine the level of agreement between two or more raters when the method of assessment, known as the response variable, is measured on a categorical scale. The Kappa results are interpreted as follows: values ≤ 0 indicate no agreement, values from 0.01 to 0.20 indicate none to slight agreement, values from 0.21 to 0.40 indicate fair agreement, values from 0.41 to 0.60 indicate moderate agreement, values from 0.61 to 0.80 indicate substantial agreement and values from 0.81 to 1.00 indicate almost perfect agreement.

A case was regarded as PD-L1 positive or -negative if more than 50% of the participants designated it as positive or negative, respectively. The consensus score was considered a majority score if 67% or more of the participants agreed on the categorisation. If all participants agreed (100%), this was regarded as absolute agreement (AA). The cases with agreement less than 67% and above 50% were considered challenging. In cases of no agreement (50% or less), a case was considered as PD-L1-positive or -negative based on the consensus of the experienced pathologists only.

Scatter plots were used to visualise percentage PD-L1 scores, and the strength of the relationship between scores was expressed as a squared correlation coefficient (R2). All analyses were supervised by an expert in pathology informatics (PL).

An outline of the study methodology is shown in Figure 1.

## 3. Results

### 3.1. Cohort Characteristics

A total of 100 breast cancers were assessed, comprising 29/93 (33.3%) grade 2 and 62/93 (66%) grade 3 cases, while 7 cases had missing grades. The patient ages ranged from 42 to 59 years, with a median of 49 years. Fifty-eight carcinomas were triple-negative, 28 were luminal and 14 were Her2-positive. All cases were evaluated independently by twelve pathologists, including nine specialist consultant breast pathologists, of whom six had 1–3 years’ experience in PD-L1 scoring, as per the survey responses. All scorers had significant experience in breast pathology reporting, and six had experience in scoring PD-L1 SP142 in TNBC in routine practice. A consultant Biomedical Scientist and two trainee pathologists were among the scorers. Ten pathologists scored round two, following a washout period. The overall percentage of PD-L1 positivity for all of the breast cancer molecular subtypes was 36–38%, and the highest was for TNBC (55%) (Table 1).

### 3.2. Inter-Observer Agreement and Pathologist Experience

The Kappa of the inter-observer agreement between the participants in classifying cases as PD-L1-positive vs. -negative and the number of cases with absolute agreement (AA) for rounds one and two were calculated. The overall agreement was substantial (Kappa 0.654 and 0.655 for the first and second rounds, respectively) (Table 2).

There was absolute agreement on scoring cases as either positive or negative in 52 cases in the first round (Figure 2A–D). This increased to 60 cases in the second round (Table 3). A further 42 and 32 cases achieved majority agreement in the first and second rounds, respectively. Overall, the Kappa value, for all cases, was similar between experienced pathologists and those without considerable experience in PD-L1 reporting. However, it was higher for expert pathologists scoring PD-L1 in the TNBC group (0.6 vs. 0.568 in the second round) (Table 4).

### 3.3. Concordance of PD-L1 Percentage Expression

When the median percentage of PD-L1 expression was considered, the expert pathologists had a higher and tighter concordance compared with the non-experts (R^2^ = 0.920 vs. 0.89). The overall concordance was excellent (R^2^ = 0.935). The distribution of the percentage scoring among all raters (those with and without experience in PD-L1 routine reporting) in both rounds is shown in Figure 3A–E.

### 3.4. Reasons for Discordance

Ten cases were regarded as challenging, with low (<67–>50%) or no agreement (<50%), most of which (8/10) were of the TNBC phenotype (Table 5). All ten cases had a low PD-L1 score, with a median range of 0.5–1%, highlighting the difficulties in classifying cases close to the cut-off value of 1%. Four cases were challenging in both rounds, indicating of the innate difficulty of the cases, regardless of the pathologists’ expertise in PD-L1 scoring. We analysed the reasons for the difficulties in scoring those cases by reviewing the digital images and referring to pathologists’ comments on scoring. Those cases were reassuringly recognised as difficult by most scorers due to the nature of the tumour and/or technical issues. Reasons for discordance included uncertainty as to the presence/extent of the in situ carcinoma, a small amount of invasive carcinoma, positive staining around the normal mammary epithelium, very focal staining, background staining and staining within areas of necrosis (Figure 2E–H).

It is of note that the consensus in scoring those challenging cases among the expert pathologists ranged from no agreement (50%) to absolute agreement (100%), with a higher percentage of the former in the first round (3/10; 30%) compared to the second (1/10; 10%). For the experts, concordance improved in the second round, with all but one case showing absolute agreement. Strong to almost perfect agreement among the experts was seen in 6/10 of those challenging cases, in both rounds. For the non-experts, the proportion of cases with low or no agreement was higher than that for the experts and increased from 30% in the first round to 50% in the second round (Table 5).

### 3.5. Inter- and Intra-Observer Agreement

Cohen’s Kappa was calculated to assess the inter-observer agreement between each of the scoring pathologists and the intra-observer agreement for each scorer across the two rounds. Inter-observer agreement was, overall, moderate (0.5) to substantial (0.75) (Table 6). The highest Kappa values for inter-observer agreement were 0.871 (in the first round) and 0.88 (in the second round), while the lowest values were moderate: 0.475 (in the first round) and 0.498 (in the second round). The intra-observer agreement was substantial to almost perfect, ranging from 0.667 to 0.956 (Table 6).

### 3.6. Intraclass Correlation Coefficient (ICC)

ICC was used to assess the reliability of scoring between different groups of raters using the median percentage expression. The ICC for different groups (all scorers, experienced scorers and non-experienced scorers) ranged from moderate (0.5–0.75) to excellent (>0.9), with the predominance of the latter (Table 7). The highest ICC was 0.974 (between all scorers in first round and experienced ones in the second round), while the lowest value was 0.619 (between the non-experienced in the first round and the experienced in the second round).

### 3.7. Intra-Observer Agreement and Scoring Reliability in Relation to Pathologists’ Experience

All scorers had significant experience in breast pathology reporting, but only six scored PD-L1 SP142 in breast cancer in routine practice. The experience in PD-L1 reporting did not appear to affect the intra-observer agreement, with all scorers showing substantial or almost perfect agreement. On the other hand, the intra-observer reliability in the percentage assessment of PD-L1 expression was higher for experienced pathologists compared with non-experienced pathologists (Table 8).

## 4. Discussion

We present comprehensive data of a large PD-L1 concordance cohort, scored twice by pathologists from eight institutions, representing three countries. Our data show reassuring inter- and intra-observer agreements, which were the highest among experts, and highlight cancers with low levels of PD-L1 expression as the most challenging in classifying as either PD-L1-positive or -negative.

Unlike standard diagnostic and prognostic markers for breast cancer, SP142 PD-L1 immunohistochemistry is assessed in the immune micro-environment of breast cancer and not in the neoplastic cells themselves. PD-L1 expression in foci of ductal carcinoma in situ (DCIS), necrotic debris, normal mammary tissue and normal nodal tissue is excluded. Therefore, experience in both tumour morphology and PD-L1 assessment is required and may affect the reproducibility of scoring.

Few studies, summarised in Table 9, have addressed the consistency of PD-L1 reporting among pathologists. A prospective multi-institutional study showed the poor reproducibility of PD-L1 scoring, with pathologists disagreeing on the classification of cases as PD-L1-positive or -negative in over half of the scored cases, and the with complete agreement of SP-142 scoring in only 38% of cases [21]. In a cohort of 426 tumours of Chinese women, the concordance between two pathologists in PDL-1 scoring was 78.2%, with a Kappa value of 0.567, and 61.4% in primary tumours and nodal metastasis, respectively, indicating moderate agreement [22]. Using “Observers Needed to Evaluate Subjective Tests” (ONEST), Reisenbichler et al. [21] reported a decreased overall percentage agreement with the increase in the number of pathologists assessing each case, with the lowest concordance at eight pathologists or more. Another study of 79 PD-L1 SP142-stained breast cancers scored by experienced breast pathologists at the Memorial Sloan-Kettering Cancer Centre revealed strong agreement [23]. Our data, based on a larger cohort of TNBC cases, confirm the substantial agreement and show that concordance was higher among experts than among those with no experience in reporting PD-L1. More importantly, the agreement among experts was observed as substantial to perfect in those challenging cases, and those experts showed a much higher consistency in reporting challenging, low-expressing TNBC, a finding that is relevant to clinical practice. This is in accordance with findings in other biomarkers [24] and reflects the importance of testing at regional institutions with quality-assured protocols and experienced scorers and the value of discussing/referring difficult/equivocal cases to expert pathologists for their opinions.

While several antibodies/assays for PD-L1 assessment are available (e.g., 22C3, 28-8, SP142, SP263 and 73-10), the VENTANA Roche SP142 assay is the only companion FDA- and CE-IVD (European Commission in vitro diagnostics)-approved test for atezolizumab therapy. An expert round table in 2019 [25] recommended the assay as the only approved companion diagnostic for selecting patients for immunotherapy and recommended using the primary tumour samples, where available, over metastases for assessment. In the UK, atezolizumab plus chemotherapy, and its companion diagnostic assay, were granted approval by the National Institute of Health and Care Excellence (NICE) for the treatment of locally advanced/metastatic PD-L1-positive TNBC. More recently, pembrolizumab plus chemotherapy has been approved for the same indication for PD-L1-positive TNBC using the companion diagnostic Agilent 22C3 assay.

In this study, we assessed both the inter- and intra-observer concordance among the participating pathologists. It is notable that the intra-observer concordance was high (0.667 to 0.956) among both expert and non-expert pathologists in PD-L1 scoring, indicating that pathologists are likely to stick to their parameters on scoring. When the median percentage of PD-1 expression was compared among the raters, the highest ICC (0.974) was achieved among experienced raters in the second round. We observed the lowest concordance value of 0.619 when comparing non-experienced to experienced scorers. Similarly, a higher concordance among those experienced in PD-L1 scoring (93.3%) compared with non-experts (81.5%) was previously reported by Pang et al. [26].

While, overall, there was a high concordance among pathologists in PD-L1 SP142 scoring, some cases were challenging to score. Those cases comprised 6–8% of all cases and generally showed very low levels of expression spanning the threshold for positivity. These may represent a so called “borderline category” where expression cannot readily be designated into a clear-cut positive or negative status. Ideally, information on the tumour response to immunotherapy should determine how those cases should be classified. It is of interest that expert pathologists, who routinely reported PD-L1 in breast cancer, showed substantial concordance in scoring those difficult cases. We therefore recommend that those cases of very low expression (i.e., close to the 1% cut-off value) are scored by an expert pathologist either via double-reporting or via a second opinion referral.

**Table 9 cancers-15-01511-t009:** Summary of studies evaluating the SP142 PD-L1 concordance of scoring.

Reference	Number of Cases (Type)	Clone(s)	SP142 Scoring Method	Scorers	Inter-Observer Agreement	Intra-Observer Agreement
Downes et al. 2020 [19]	30 surgical excisions TMAs	22C3, SP142, E1L3N	IC ≥ 1%	3 pathologists	Kappa for IC1%: 0.668	1 month washout period. Kappa = 0.798
Noske et al. [13]	30 (resections)	SP263, SP142, 22C3, 28–8	IC ≥ 1%	7 trained + one Ventana SP142 expert for SP142 only	ICC for SP142: 0.805 (0.710–0.887)	Not tested
Dennis et al. (abstract) [14]	28 test sets through the Roche International Training Programme	SP142	IC ≥ 1%	432 (trained multiple institutions), from several countries	OPA: was 98.2%, with PPA of 99.4% and NPA of 96.6%.	Not tested
Hoda et al. [23]	75 (cores and excision), primary and metastases	SP142	IC ≥ 1%	8 experienced (single institution)	Kappa 0.727	Not tested
Reisenbichler et al. 2021 [21]	68 cases for SP142 and 67 cases for SP263	SP142, SP263	IC ≥ 1% & % expression for cases scored as positive only	19 randomly selected pathologists from 14 US institutions; breast pathologists, with few non-breast pathologists. Experience in reporting PD-L1 not stated	Complete agreement for SP142 categorisation into positive vs. negative in 38%. Agreement decreased with the increasing number of scorers, reaching a low plateau of 0.41 at eight scorers or more	Not tested
Pang et al. [26]	60 TNBC TMAs	VENTANA SP142, DAKO 22C3	IC ≥ 1%	10 pathologists including 5 PD-L1 who were naïve and 5 who passed a proficiency test	93.3% for experts; 81.5% for non-experts.	Tested after a 1 h training video and an overnight washout period. OPA increased from 81.5% to 85.7% for non-experts after video training. OPA was 96.3% for experts.
Van Bockstal et al. 2021 [15]	49 metastatic TNBC (biopsies and resections)	VENTANA SP142	IC ≥ 1%	10 pathologists; all passed a proficiency test	Substantial variability at the individual patient level. In 20% of cases, chance of allocation to treatment was random, with a 50–50 split among pathologists in designating as PD-L1-positive or -negative	Not tested
Ahn et al. 2021 [27]	30 surgical excisions	SP142, SP263, 22C3 and E1L3N	ICs and TCs were scored in both continuous scores (0–100%) and five categorical scores (<1%, 1–4%, 5–9%, 10–49% and ≥50%).	10 pathologists with no special training, of whom 6 underwent Ventana Roche training	80.7% inter-observer agreement at a 1% cut-off value	Proportion of cases with identical scoring at a 1% IC cut-off value increased from 40% to 70.0% after training
Abreu et al. 2022 (Conference abstract) [28]	168 in tissue microarrays	22C3 and SP142	Not stated	4 pathologists including 2 breast pathologists and 2 surgical pathologists with no specific PD-L1 training	Overall concordance for SP142 was 64.8%; overall κ = 0.331, with κ = 0.420 for breast pathologists and κ = 0.285 for general pathologists	Not tested
Chen et al. 2022 [22]	426 primary and metastatic surgical excisions	SP142	IC ≥ 1%	Two experienced pathologists	78.2% concordance; κ = 0.567	Not tested
Current study	100 (cores), primary breast cancer	SP142	IC ≥ 1% & % expression for all cases; two rounds of scoring separated by a 3-month washout period	12 experienced breast pathologists from 8 institutions in the UK, Ireland and Belgium. All passed a proficiency test.	Absolute agreement was substantial in 52% and 60% of cases in the first and second rounds, with Kappa values of 0.654 and 0.655 for the first and second rounds, respectively. Higher concordance among experts, particularly in TNBC and challenging cases.	Tested after 3 months of a washout period. Almost perfect agreement regardless of pathologists’ PD-L1 experience

Similar challenges in PD-1 scoring have been highlighted in carcinomas in other tissues. For example, the concordance between the assays used for PD-L1 assessment in head and neck squamous cell carcinoma (HNSCC) was fair to moderate, with a tendency for the SP142 assay to better stain the immune cells [29]. Furthermore, using 3 PD-L1 tests for HNSCC tissue microarrays (standard SP263, standard 22C3 and in-house-developed 22C3), significant differences were found among the three tests using clinically relevant cut-off values, i.e., ≥20 and ≥50%, for the combined positive score (CPS) and Tumour positive score (TPS). Intra-tumour heterogeneity was generally higher when CPS was used [30]. On the other hand, Cerbelli et al. showed a high concordance between the 22C3 PharmDx assay and the SP263 assay on 43 whole sections of HNSCC [31]. The data collectively highlight the challenges in PD-L1 assessment in various cancers, including the differences in the results between the available antibody clones and staining platforms.

Our data also confirm previous studies showing the highest proportion of PD-L1 positivity in TNBC [32]. PD-L1 was previously shown to be associated with higher tumour grades and higher pCR rates. Low levels of expression were associated with shorter recurrence-free survival (RFS), including following subtype adjustment [32].

The current study and previous lessons from the IMpassion trial [33] shed some light on issues related to the immunohistochemical assessment of PD-L1 in breast cancer tissue. The strengths of the study include the large cohort of cases, the inclusion of 12 pathologists from three countries, the inclusion of both expert and non-expert assessors, the robust design, with the assessment of inter- and intra-observer concordance in two rounds, and the detailed statistical analysis. The digital analysis of whole slide images, rather than scoring glass slides, may be a weakness for pathologists who are not used to digital reporting. More recently, the use of digital image analysis algorithms and/or artificial intelligence (AI) has been proposed for PD-L1 scoring in various solid tumours [34]. Going forward, this is an exciting and promising endeavour that requires thorough validation in comparison to the gold-standard pathologist scoring before implementation and the determination of whether those algorithms are superior to manual scoring in identifying responders to immune therapy. Currently, PD-L1 Artificial Intelligence (AI) scoring in breast cancer is limited to research studies and has not been validated for routine clinical use.

## 5. Conclusions

In summary, we present a detailed analysis of 12 pathologists who scored 100 digitally scanned breast cancer slides for PD-L using the Ventana SP142 assay in two rounds separated by a washout period. Absolute (100%) agreement was substantial in 52% and 60% of cases in the first and second rounds, with Kappa values of 0.654 and 0.655 for rounds one and two, respectively. We provide reassuring evidence of a high concordance of PD-L1 reporting among pathologists, the highest being among experts and in reporting challenging, low-expressing TNBC. The intra-observer agreement was substantial for all raters. Despite experience and the adherence to current reporting guidelines, there remains a minority of tumours (6–8%) that are challenging to assign to either a positive or negative category. Those are PD-L1 low-expressing and/or heterogeneous tumours that suffer from the least concordance among pathologists. Consensus scoring and referrals for expert opinions should be considered in those cases. If uncertainly persists, this should be recognised and well communicated to clinicians in the context of a multidisciplinary approach. For inconclusive cases, testing on another tumour sample and/or using another assay (e.g., the DAKO 22C3 assay for selecting patients for pembrolizumab therapy) could be performed.

Pathologists’ training and experience are paramount in evaluating PD-L1 expression and selecting patients for immune checkpoint anti-PD-L1 inhibitors. Further work on refining the criteria for scoring, pathologists’ training and assessing pathologist concordance is needed. This will ensure the accurate classification of tumours into a positive or negative category and, hence, the accurate selection of patients for atezolizumab therapy.

This study also shows that digital pathology is a useful tool that allows for the instantaneous sharing of high-quality whole slide scans with colleagues. This is particularly helpful for consensus scoring and/or seeking expert opinions.

## Figures and Tables

**Figure 1 cancers-15-01511-f001:**
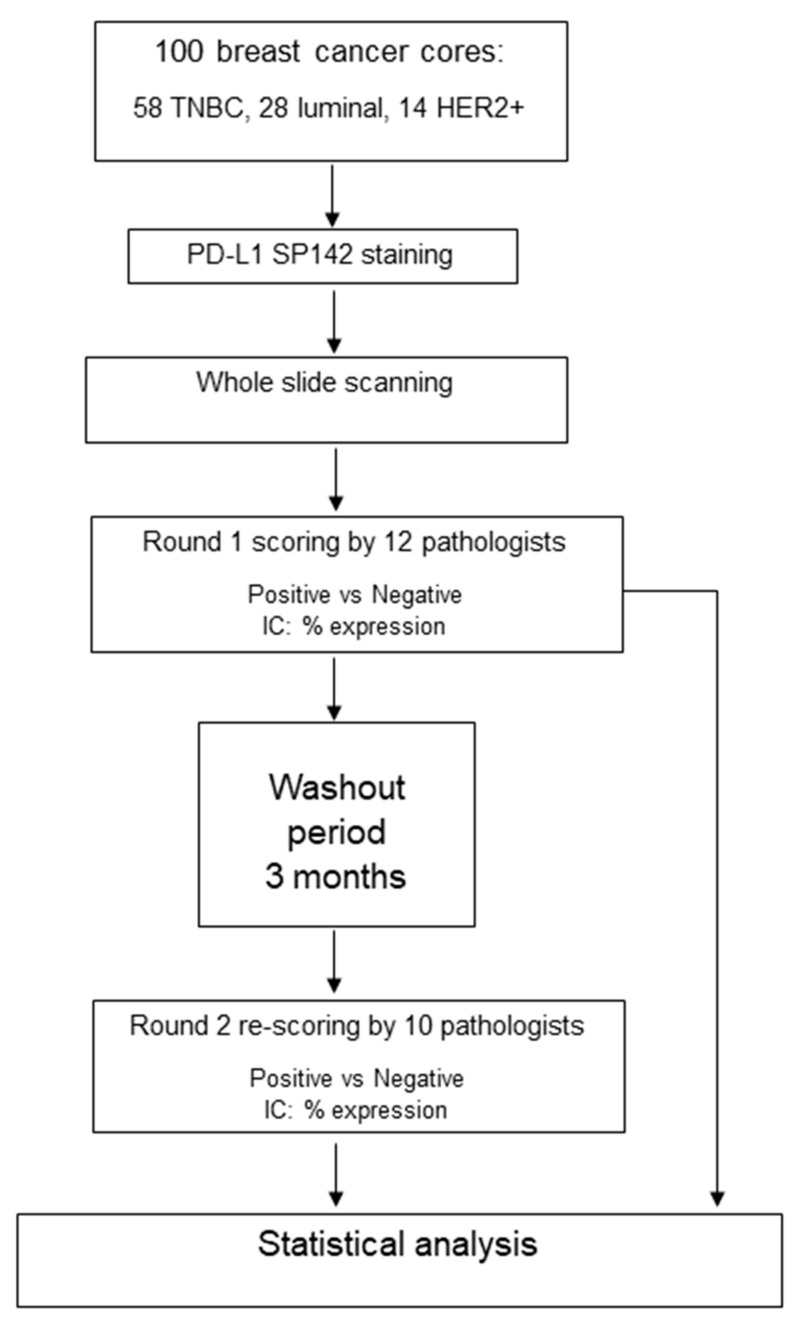
A flowchart showing the outline of the study.

**Figure 2 cancers-15-01511-f002:**
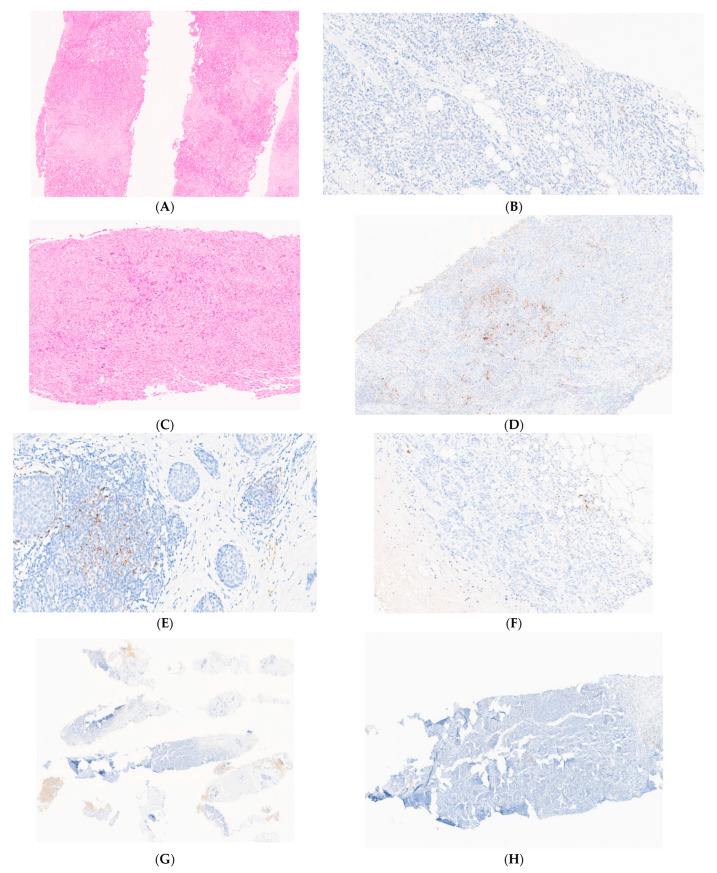
Examples of breast cancer PD-L1 scores using the Ventana SP142 assay and challenging cases: (**A**) H&E staining of three cores of invasive no-special-type carcinoma (×50); (**B**) Only focal PD-L1 staining is noted (<1%), and the case was classified as PD-L- negative (×100); (**C**) H&E staining of one core of invasive no-special-type carcinoma (×100); (**D**) Higher magnification of PD-L1 immunohistochemistry shows strong positivity with absolute agreement among all scorers in both rounds (×100); (**E**) Challenging case due to uncertainty as to whether the PD-L1 staining observed is associated with in situ or invasive carcinoma (×100); (**F**) Challenging cases showing low levels of expression. Experts’ consensus was to designate the case as PD-L1-negative (×100); (**G**) A case with no consensus in either round. Low-power view showing areas of tumour necrosis and background staining (×15); (**H**) Higher magnification showing focal expression in tumour stroma and adjacent to an area of necrosis (×50). This case was also challenging for experts; in the first round, it showed low agreement (60% as negative), and in the second round, it showed no agreement (50%).

**Figure 3 cancers-15-01511-f003:**
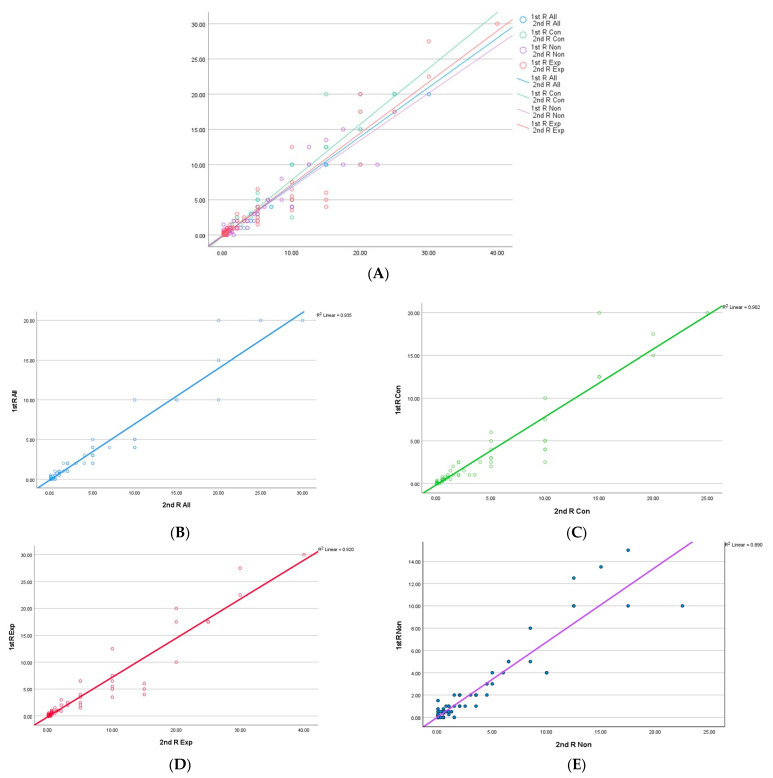
Scatter plot showing the distribution of the median percentage of PD-L1 in the four groups; experienced consultants (Exp; red line), non-experienced consultants (Con; green line) and all (All; blue line) participants. (**A**) The distribution of percentage scores among all scorers in both rounds; (**B**) All pathologists’ (including non-experts and trainees) percentage scores in both rounds (R^2^ = 0.935); (**C**) All consultants’ (including experts and non-experts) percentage scores in both rounds (R^2^ = 0.902); (**D**) Experienced consultants’ percentage scores in both rounds (R^2^ = 0.920); (**E**) Non-experienced pathologists’ (including trainees) percentage scores in both rounds (R^2^ = 0.89).

**Table 1 cancers-15-01511-t001:** Frequency of PD-L1 positivity in breast cancer in both rounds, stratified by the molecular type.

	No.	First Round		Second Round	
		Positive (%)	Negative (%)	Positive (%)	Negative (%)
TNBC	58	32 (55%)	26 (45%)	32 (55%)	26 (45%)
Median (range)		4 (0.75–30)	0 (0–1)	5 (0.5–30)	0 (0–1)
Luminal	28	4 (14%)	24 (86%)	4 (14%)	24 (86%)
Median (range)		2 (1–4)	0 (0–0.75)	3.5 (1.5–5)	0 (0–0.5)
Her2-positive	14	2 (14%)	12 (86%)	1 (7%)	13 (93%)
Median (range)		5.5 (1–10)	0 (0–0.5)	10	0 (0–0.5)
Total	100	38 (38%)	62 (62%)	36 (36%)	64 (64%)
Median (range)		2 (0.75–30)	0 (0–1)	5 (0.5–30)	0 (0–1)

**Table 2 cancers-15-01511-t002:** Absolute agreement in scoring among raters in the two rounds.

	Raters	P1	P2	P3	P4 ^c^	P5 ^c^	P6 ^c^	P7 ^e^	P8 ^e^	P9 ^e^	P10 ^e^	P11 ^e^	P12 ^e^
First Round	Neg	62	64	61	58	63	68	75	51	64	67	63	63
	Pos	38	35	31	41	37	32	25	49	36	33	37	34
	Total	100	99	92	99	100	100	100	100	100	100	100	97
	Kappa	0.654											
	AA	52/100 cases; 36 scored negative and 16 scored positive											
Second Round	Neg	60	64		64	46	62		52	72	69	69	66
	Pos	40	35		35	54	37		48	28	31	31	30
	Total	100	99		99	100	100		100	100	100	100	97
	Kappa	0.655											
	AA	60/100 cases; 40 scored negative and 20 scored positive											

^e^ Experienced consultant, ^c^ Consultant, (AA) Absolute agreement. P3 and P7 did not score the second round.

**Table 3 cancers-15-01511-t003:** Agreement categories in the first and second scoring rounds.

Round		Consensus (Agreement)			No Agreement
		Majority		Challenging/Low Agreement	≤50%
		100% (AA)	67–99%	<67–>50%	
First	Negative	36	24	2	0
	Positive	16	18	4	0
	Total	52	42	6	0
		94		6	0
		100			0
Second	Negative	40	20	2	2
	Positive	20	12	4	0
	Total	60	32	6	2
		92		6	2
		98			2

**Table 4 cancers-15-01511-t004:** Fleiss Kappa of agreement between the pathologists in both rounds.

	Fleiss Kappa First Round			Fleiss Kappa Second Round		
	Scoring Categories			Scoring Categories		
	Overall (TNBC)	NEG	POS	Overall (TNBC)	NEG	POS
All	0.654 (0.61)	0.660	0.678	0.655 (0.602)	0.656	0.669
Consultants	0.663 (0.616)	0.664	0.673	0.633 (0.568)	0.636	0.650
Experienced	0.659 (0.642)	0.661	0.672	0.674 (0.600)	0.677	0.695

**Table 5 cancers-15-01511-t005:** Distribution of ten challenging (low concordance and no agreement) cases in both rounds.

		FIRST ROUND									SECOND ROUND								
Type		ALL (12)	PD-L1 Status	M	NON (6)	PD-L1 Status	M	EXP (6)	PD-L1 Status	M	All (10)	PD-L1 Status	M	Non (5)	PD-L1 Status	M	EXP (5)	PD-L1 Status	M
TNBC		6/11 (55%)	−	0.5	3/6 (50%)		0.75	3/5 (60%)	−	0.5	6/9 (67%)	−	0.5	4/5 (80%)	−	0	2/4 (50%)		1
Her2		7/12 (58%)	−	1	4/6 (67%)	−	1	3/6 (50%)		0.75	7/10 (70%)	−	0.5	3/5 (60%)	*+*	1	5/5 (100%)	−	0.5
TNBC		7/12 (58%)	−	0.5	4/6 (67%)	−	0.25	3/6 (50%)		0.75	6/10 (60%)	+	1	4/5 (80%)	*+*	1	3/5 (60%)	−	0.5
TNBC		7/12 (58%)	−	1	3/6 (50%)		1	4/6 (67%)	−	1	6/10 (60%)	−	1	3/5 (60%)	*+*	1	4/5 (80%)	−	0.75
TNBC		7/12 (58%)	+	1	4/6 (67%)	+	1	3/6 (50%)		0.75	5/9 (56%)	+	1	2/4 (50%)		2	3/5 (60%)	+	0.75
TNBC		7/12 (58%)	−	1	4/6 (67%)	*+*	1	5/6 (83%)	−	0.5	5/10 (50%)		1	4/5 (80%)	*+*	1.5	4/5 (80%)	−	0.75
TNBC		9/12 (75%)	+	1	4/6 (67%)	+	1	5/6 (83%)	+	1	6/10 (60%)	+	2	4/5 (80%)	*+*	2.5	3/5 (60%)	−	0.5
TNBC		8/12 (67%)	+	1	3/6 (50%)		0.5	5/6 (83%)	+	1	6/10 (60%)	−	1	3/5 (60%)	−	0.5	4/5 (80%)	+	1.5
TNBC		11/12 (83%)	−	0.5	6/6 (100%)	−	0.5	5/6 (83%)	−	0.5	6/10 (60%)	−	0.5	3/5 (60%)	*+*	1	4/5 (80%)	−	0.5
Lum		8/12 (75%)	+	1	4/6 (67%)	+	1	4/6 (67%)	+	1	5/10 (50%)		1.5	4/5 (80%)	*+*	3.5	4/5 (80%)	−	0.5
**AGREEMENT**	** No **	** 0 **			** 3/10; 30% **			** 3/10; 30% **			** 2/10; 20% **			** 1/10; 10% **			** 1/10; 10% **		
	** Low **	** 6/10; 60% **			** 0 **			** 1/10; 10% **			** 6/10; 60% **			** 4/10; 40% **			** 3/10; 30% **		
	** High **	** 4/10; 40% **			** 7/10; 10% **			** 6/10; 60% **			** 2/10; 20% **			** 5/10; 50% **			** 6/10; 60% **		

PD-L1 status: negative (−) or positive (+); (M) Median percentage score; (ALL) refers to all scorers, which were then divided into (NON) Non-expert participants in PD-L1 scoring and (EXP) Expert pathologists. Cases in green represent challenging cases in the first round, those in orange represent challenging ones in both rounds and those in grey represent challenging and no-agreement cases in the second round only. The yellow cases represent contradictory agreement of non-experts in relation to the experts’ agreement, and the blue cases represent cases with no agreement (50%). Agreement levels are categorised into no-agreement (50%), low-agreement (>50%–<67%) and high-agreement (>67%).

**Table 6 cancers-15-01511-t006:** Inter- and intra-observer agreement for all scoring pathologists.

	Consensus 1	P1	P2	P3	P4	P5	P6	P7	P8	P9	P10	P11	P12
**Consensus 2**	** 0.912 **	**0.851**	**0.78**		**0.823**	**0.629**	**0.715**		**0.737**	**0.747**	**0.819**	**0.865**	**0.884**
P1	*0.892*	** 0.832 **	**0.766**		**0.679**	**0.724**	**0.787**		**0.758**	**0.649**	**0.719**	**0.762**	**0.733**
P2	*0.765*	*0.747*	** 0.722 **		**0.735**	**0.551**	**0.695**		**0.575**	**0.562**	**0.729**	**0.774**	**0.745**
P3	*0.786*	*0.768*	*0.607*										
P4	*0.823*	*0.762*	*0.654*	*0.641*	** 0.956 **	**0.587**	**0.669**		**0.631**	**0.606**	**0.637**	**0.728**	**0.699**
P5	*0.74*	*0.723*	*0.64*	*0.696*	*0.607*	** 0.667 **	**0.682**		**0.801**	**0.498**	**0.554**	**0.515**	**0.539**
P6	*0.798*	*0.737*	*0.741*	*0.632*	*0.617*	*0.713*	** 0.732 **		**0.632**	**0.635**	**0.688**	**0.643**	**0.656**
P7	*0.718*	*0.659*	*0.579*	*0.717*	*0.669*	*0.54*	*0.634*						
P8	*0.678*	*0.658*	*0.533*	*0.543*	*0.613*	*0.678*	*0.577*	*0.475*	** 0.94 **	**0.552**	**0.574**	**0.614**	**0.661**
P9	*0.891*	*0.871*	*0.745*	*0.764*	*0.715*	*0.72*	*0.733*	*0.744*	*0.658*	** 0.772 **	**0.784**	**0.64**	**0.826**
P10	*0.822*	*0.76*	*0.634*	*0.831*	*0.687*	*0.649*	*0.704*	*0.711*	*0.597*	*0.801*	** 0.862 **	**0.762**	**0.88**
P11	*0.87*	*0.808*	*0.724*	*0.649*	*0.738*	*0.743*	*0.801*	*0.586*	*0.638*	*0.763*	*0.737*	** 0.778 **	**0.784**
P12	*0.843*	*0.735*	*0.646*	*0.758*	*0.708*	*0.643*	*0.7*	*0.687*	*0.563*	*0.732*	*0.749*	*0.732*	** 0.906 **

Figures in italics below the equatorial bordered cells represent the values of the first round, while those in bold represent the second round. The equatorial bordered cells (in bold red font) represent the intra-observer agreement for each participant, as scored in both rounds. Cell shading colours reflect the level of agreement as follows; light green for almost perfect agreement (0.81–1), orange for substantial agreement (0.61–0.8) and light red for moderate agreement (0.41–0.6).

**Table 7 cancers-15-01511-t007:** Intraclass correlation coefficient for all groups of pathologists.

	ALL-1	EXP-1	NON-1	ALL-2	EXP-2	NON-2
ALL-1		0.907	0.931	0.906	0.768	0.932
EXP-1	** *0.915* **		0.772	0.974	0.913	0.919
NON-1	** *0.933* **	** *0.788* **		0.781	0.619	0.876
ALL-2	** *0.919* **	** *0.974* **	** *0.804* **		0.911	0.946
EXP-2	** *0.798* **	** *0.923* **	** *0.655* **	** *0.919* **		0.792
NON-2	** *0.936* **	** *0.920* **	** *0.891* **	** *0.949* **	** *0.808* **	

Figures in bold/italics below the equatorial grey cells represent values of ICC calculated according to the consistency of assessment, while values below equatorial cells represent ICC calculated according to absolute agreement. (ALL): All scorers’ median percentage; (EXP): Experienced scorers’ median percentage; (NON): Non-experienced scorers’ median percentage. Cells’ shading colours reflect the level of reliability as follows: Red for moderate reliability (0.5–0.75); Orange for good reliability (0.75–0.9); Green for excellent reliability (greater than 0.9).

**Table 8 cancers-15-01511-t008:** Intra-observer concordance based on pathologists’ experience in scoring PD-L1.

Rater	Position	Experience as a Breast Reporting Pathologist (years)	Experience in SP142 PD-L1 Reporting (years)	Previous Training in SP142 PD-L1 Reporting (Provider)	Intra-Observer Agreement (Cohen’s Kappa/Level of Agreement)	Intra-Observer Reliability (ICC/Level of Reliability)
P1	Trainee Pathologist	12	0	Roche	0.832/Almost perfect	0.826/Good
P2		12	0	Roche	0.722/Substantial	0.525/Moderate
P3	Consultant Scientist	N/A	0	Roche	N/A/N/A	N/A/N/A
P4	Consultant Pathologist	20	0	N/S	0.956/Almost perfect	0.852/Good
P5		21	0	Roche	0.667/Substantial	N/A/N/A
P6		25	0	None	0.732/Substantial	0.770/Good
P7		25	3	Roche	N/A/N/A	N/A/N/A
P8		29	1	Roche	0.94/Almost perfect	0.935/Excellent
P9		10	2	Roche	0.772/Substantial	0.933/Excellent
P10		25	2	Roche	0.862/Almost perfect	0.920/Excellent
P11		30	3	Local	0.778/Substantial	0.756/Good
P12		22	2	Roche	0.906/Almost perfect	0.929/Excellent

(N/S) Not stated; (N/A) Not applicable.

## Data Availability

Full data are available from the corresponding author upon reasonable request. Digital slides are password-protected and available at the University of Birmingham platform: https://eslidepath.bham.ac.uk, Last accessed 23 February 2023.

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
