# Peer review of "Inter- and Intra-Observer Agreement of PD-L1 SP142 Scoring in Breast Carcinoma—A Large Multi-Institutional International Study"

_cancers, 2023, doi:10.3390/cancers15051511_

Round 1

Reviewer 1 Report

Zaakouk et al. studied inter- and intra-observer concordance of pathologists' scorings of PD-L1 SP142 assay in 100 invasive breast carcinomas. Twelve pathologists from three European countries participated in the study, including nine experts and three non-experts. 

  1. The authors need to rewrite parts of the text and simplify the tables for better readability.  
  2. Please check the details carefully. There are many spelling and punctuation mistakes. For example, PD-L1 and PDL-1 were not used consistently in the manuscript. There are places missing parenthesis, spaces, and misuse of punctuation. References were cited twice in the discussion. 
  3. Suggestion to add an outline of the study. 
  4. The discussion needs to be toned down. Add relevant references and limitations of the current study.
  5. Table 9, the comparison of PD-L1 concordance of scoring from different studies seems better suited for a review paper. Not sure how to compare the number of cases, as they are listed as resections, cores, cases, and TMA. 

Author Response

We thank the reviewer for their time and feedback to improve the manuscript readability. We addressed all the comments raised, point by point, below and in the manuscript. All changes have been made in blue for easy identification. We hope that the revised version meets your expectations.

Zaakouk et al. studied inter- and intra-observer concordance of pathologists' scorings of PD-L1 SP142 assay in 100 invasive breast carcinomas. Twelve pathologists from three European countries participated in the study, including nine experts and three non-experts. 

  1. The authors need to rewrite parts of the text and simplify the tables for better readability.  

The manuscript has been wholly reviewed and some sections, including tables and abbreviations re-written and simplified. The tables have been edited and better explained in the test.

  1. Please check the details carefully. There are many spelling and punctuation mistakes. For example, PD-L1 and PDL-1 were not used consistently in the manuscript. There are places missing parenthesis, spaces, and misuse of punctuation. References were cited twice in the discussion. 

The manuscript was reviewed and “PDL-1” was replaced with “PD-L1” in three places. The whole manuscript is now consistent in using “PD-L1” throughout. The formatting and punctuation of the whole manuscript has been edited and adjusted.

  1. Suggestion to add an outline of the study. 

Thank you for the suggestion. A new figure (Figure 1) outlining the study has been inserted as requested.

  1. The discussion needs to be toned down. Add relevant references and limitations of the current study.

The discussion has been rewritten with relevant references added and comparison with challenges in PD-L1 scoring in other tissues (head and neck squamous cell carcinoma as an example) has been added. Strengths and weaknesses of the study have been included. We have also added a recent study analysing the concordance between primary and metastatic TNBC assessment and intra observer concordance of two pathologists (Chen et al 2022)

  1. Table 9, the comparison of PD-L1 concordance of scoring from different studies seems better suited for a review paper. Not sure how to compare the number of cases, as they are listed as resections, cores, cases, and TMA. The data on breast cancer PDL-1 concordance of scoring in the literature is limited. We sought to present the data available so far in a table format in the discussion to improve readability and understanding. We highlighted the differences in the design of various studies to help readers interpret current data in light of previous results. We agree with the reviewer that the differences in previous studies design, nature of tissues, differences in platforms and antibodies used makes comparing studies rather challenging which is what we sought to highlight. We believe that presenting this complex data in a table format is useful and provides the readers with an overall view of the current studies in this area and allows comparison with the current study.

Reviewer 2 Report

Overall , it is clear , concise and well written manuscript, the introduction is relevant sufficient  information about previous study findings is presented to the reader to follow present study rationales and procedure  ,the authors have collected and used unique dataset with excellent efforts of 12 pathologist who tried to do their best evaluating PD-L1 SP124 stain in different types of breast cancer biopsies .However in my opinion the paper has shortcoming in regard to some text particularly the conclusion which looks short noted and insufficient  comparing with analysis that has not utilized into it's full extent , I suggest  to revise the conclusion and to add more information that relevant to the analysis of the data that been done in order to strengthen the conclusive results obtained and to be more presentable to the readers.

Author Response

We thank the review for the positive comments that are greatly received. 

Overall , it is clear , concise and well written manuscript, the introduction is relevant sufficient  information about previous study findings is presented to the reader to follow present study rationales and procedure  ,the authors have collected and used unique dataset with excellent efforts of 12 pathologist who tried to do their best evaluating PD-L1 SP124 stain in different types of breast cancer biopsies .However in my opinion the paper has shortcoming in regard to some text particularly the conclusion which looks short noted and insufficient  comparing with analysis that has not utilized into it's full extent , I suggest  to revise the conclusion and to add more information that relevant to the analysis of the data that been done in order to strengthen the conclusive results obtained and to be more presentable to the readers.

Both the discussion and conclusion sections have been extensively edited and updated. Pertinent references and discussion of the data in light of previously published literature in breast and other tissues have been added. The conclusion has been fully updated to include the most important study findings, clinical implications and future directions.

All changes in the manuscript are made in blue for easy identification. We hope that the revised manuscript meets your expectations.

Reviewer 3 Report

The paper is focused on relevant topic: inter and intra observer agreement of PD-L1 SP142 scoring in breast carcinoma. 

The manuscript is clear and presented in a well structured manner. The study is well designed. Materials and methods are described in detail. 

Results are reported clearly and appropriate. Tables and figures properly show the data. The conclusions are consistent with the evidence. 

The cited references are mostly within the last 5 years, no self-citations were found. The ethics statements and data availability statements are adequate.

Just minors:

- to improve the quality of your figures (EE + ICH)

- to discuss the issue of the interobserver as managed in other fields of pathology where clinical relevant cut off are the same, to discuss the question of clones and platforms. Would recommend to acknowledge fundamental references as follow:

Challenges facing pathologists evaluating PD-L1 in head & neck squamous cell carcinoma. Girolami I et al. J Oral Pathol Med. 2021 Oct;50(9):864-873. doi: 10.1111/jop.13220. Epub 2021 Jul 8. PMID: 34157159 

Concordance between Three PD-L1 Immunohistochemical Assays in Head and Neck Squamous Cell Carcinoma (HNSCC) in a Multicenter Study.  Guerini Rocco E et al. Diagnostics (Basel). 2022 Feb 13;12(2):477. doi: 10.3390/diagnostics12020477. PMID: 35204568 

Evaluating programmed death-ligand 1 (PD-L1) in head and neck squamous cell carcinoma: concordance between the 22C3 PharmDx assay and the SP263 assay on whole sections from a multicentre study. Cerbelli B et al. Histopathology. 2022 Jan;80(2):397-406. doi: 10.1111/his.14562. Epub 2021 Nov 11. PMID: 34496080 

Author Response

We thank the reviewer for their positive comments, greatly appreciated.

Please see below the reponse to reviewer's comments

The paper is focused on relevant topic: inter and intra observer agreement of PD-L1 SP142 scoring in breast carcinoma. 

The manuscript is clear and presented in a well structured manner. The study is well designed. Materials and methods are described in detail. 

Results are reported clearly and appropriate. Tables and figures properly show the data. The conclusions are consistent with the evidence. 

The cited references are mostly within the last 5 years, no self-citations were found. The ethics statements and data availability statements are adequate.

Just minors:

- to improve the quality of your figures (EE + ICH)

The figures have been replacedwith high quality photos and the figure legend text adjusted accordingly.

- to discuss the issue of the interobserver as managed in other fields of pathology where clinical relevant cut off are the same, to discuss the question of clones and platforms. Would recommend to acknowledge fundamental references as follow:

Challenges facing pathologists evaluating PD-L1 in head & neck squamous cell carcinoma. Girolami I et al. J Oral Pathol Med. 2021 Oct;50(9):864-873. doi: 10.1111/jop.13220. Epub 2021 Jul 8. PMID: 34157159 

Concordance between Three PD-L1 Immunohistochemical Assays in Head and Neck Squamous Cell Carcinoma (HNSCC) in a Multicenter Study.  Guerini Rocco E et al. Diagnostics (Basel). 2022 Feb 13;12(2):477. doi: 10.3390/diagnostics12020477. PMID: 35204568 

Evaluating programmed death-ligand 1 (PD-L1) in head and neck squamous cell carcinoma: concordance between the 22C3 PharmDx assay and the SP263 assay on whole sections from a multicentre study. Cerbelli B et al. Histopathology. 2022 Jan;80(2):397-406. doi: 10.1111/his.14562. Epub 2021 Nov 11. PMID: 34496080 

The challenges of PD-L1 reporting in other fields, with Head and neck squamous cell carcinoma presented as an example, have been discussed. The above references have been cited and discussed. Differences in antibodies used, staining platforms and interpretation using the clinically validated cut offs have been discussed.

All changes in the manuscript have been made in blue for easy identification. We hope that the revised manuscript meets your expectations.

Round 2

Reviewer 1 Report

The authors have done a good job in the revision of this manuscript. I appreciate the authors' efforts in making the flow chart for the study outline. All the major concerns in the previous review were satisfactorily addressed.

I noticed a few typos in the paper and wish to point them out to the authors. line 299 'PDL-1' to 'PD-L1'; line 331 and 350 'PD-1' to 'PD-L1'